# Extraction of Urban Built-Up Areas Using Nighttime Light (NTL) and Multi-Source Data: A Case Study in Dalian City, China

Xueming Li [1,2], Yishan Song [1,2,*], He Liu [1,2] and Xinyu Hou [1,2]

1  School of Geography, Liaoning Normal University, Dalian 116029, China
2  Human Settlements Research Center, Liaoning Normal University, Dalian 116029, China
*  Correspondence: songyishan0916@163.com

**Abstract:** The rapid urban development associated with China's reform and opening up has been the source of many urban problems. To understand these issues, it is necessary to have a deep understanding of the distribution of urban spatial structure. Taking the six districts of Dalian as an example, in this study, we integrated the enhanced vegetation index, points of interest, and surface temperature data into night light data. Furthermore, herein, we analyze the kernel density of the points of interest and construct three indices using image geometric mean: a human settlement index (HSI), a HSI-POI (HP) index, and a HSI-POI-LST (HPL) index. Using a support vector machine to identify the land type in Dalian's built-up area, 1000 sampling points were created for verification. Then, the threshold boundary corresponding to the highest overall accuracy of each index and kappa coefficient was selected. The relevant conclusions are as follows: As compared with the other three types of data, the HPL index constructed in this study exhibited natural and social attributes, and the built-up area extracted using this method had the highest accuracy, a high image spatial resolution, and was able to overcome the omission issues observed when using one or two data sources. In addition, this method produces richer spatial details of the actual built-up area and provides more choices for assessing small-scale urban built-up areas in future research.

**Keywords:** built-up area extraction; POI; HSI; LST; Dalian city

## 1. Introduction

During the reform and opening up from 1978 to the end of 2020, China's urbanization rate increased from 17.9% to 63.89%. Although the gap between China's urbanization and that of developed countries is gradually narrowing, rapid urbanization has brought many challenges, including deteriorating air quality [1], water pollution [2], the heat island effect [3,4], traffic congestion [5], and ecological land destruction [6,7]. To assess this, a deep understanding of urban spatial structure is required. The spatial distribution of urban built-up areas is of great significance for scientific environmental management and sustainable economic development. At this stage, there is no unified definition of a built-up area; however, the "Basic Data Standard for Urban Planning" (GB/T50280-98) defines the built-up area as the area within a city's administrative district that is developed and constructed, and where municipal utilities and public facilities are available. The official statistics of built-up areas are updated slowly. In addition, built-up areas' statistical data are limited to numbers and no spatial location information is published. This makes the study of built-up areas in cities very difficult, despite it being a hot topic [8].

Nighttime light (NTL) remote sensing imagery relies on data that are composed of the light emitted from objects such as buildings on the ground captured over time [9]. Previous studies have been limited to a single source of night light to identify built-up areas and to delineate built-up areas using transforming thresholds. For example, Sutton et al. [10] relies on personal subjective experience to set thresholds to extract the boundaries of built-up

areas in the United States. IMHOFF et al. [11] uses mutation detection to extract urban built-up area boundaries in multiple regions of the United States. Moreover, the Milesi et al. [12] statistical data comparison method compares data with the officially released urban built-up area by setting a series of thresholds: when the DN value is 50, the gap between the extracted urban built-up area and the statistical data is the smallest. Shu et al. [13] used the personal empirical method, mutation detection, and the statistical data comparison method to extract the NTL threshold to identify the built-up area of Shanghai. It is relatively easy to use a single data source to identify built-up areas; however, as a result of the problems of low resolution, light saturation, and the flowering effect when using a single night light data source, the scope of the urban built-up area is always estimated to be larger than the actual built-up area [14].

In order to solve the problems associated with identifying built-up areas using single nighttime lighting data sources, some scholars have tried to introduce remote sensing image products into nighttime lighting data to improve the accuracy of built-up area boundary extraction and reduce the blooming effect of nighttime lighting data. In order to obtain more accurate results, night light data are often combined with terrestrial high-resolution remote sensing satellite imagery. For example, Deng et al. [15] used nighttime light data and Gaofen-1 remote sensing images to obtain a high-precision built-up area boundary for Yucheng County, Henan Province, using a decision tree classification algorithm. On the basis of Sentinel-2A high-definition remote sensing images, Liu et al. [16] proposed combining building correction normalized remote sensing images with night light data. In addition, Song et al. [16] combined Google Earth high-definition remote sensing images with NTL to extract built-up area boundaries. Pandey et al. [17] combined the effects of SPOT-VGT remote sensing with nighttime light data to extract built-up areas in selected cities in India. However, in some cities, there are mixed areas containing bare land and built-up areas, and high resolution images of these spaces exhibit similar spectral characteristics. This causes the division of built-up areas to be inaccurate. In addition, data collection began recently, thus it is impossible to analyze spatiotemporal changes in urban built-up areas over a long period and data are expensive to obtain. Low- and medium-resolution remote sensing images are widely used in night-light data due to their low acquisition cost, such as Landsat products. Landsat can be used to reflect natural factors, while vegetation abundance and NTL characteristics are negatively correlated. Thus, they are integrated into night light data. Lu et al. [18] was the first to construct the human settlements index (HSI) from night light data and normalized vegetation index data. In addition, Liu et al. [19] based on nighttime lighting data using the HSI method, compared statistical data to the simple threshold method to obtain the highest accuracy of urban built-up areas. Zhang et al. [20] used the normalized difference vegetation index (NDVI) and NTL to construct VANUI to reduce the saturation of night light data. However, NDVI data are limited for urban bare soil and urban area identification. Liu et al. [21] used normalized difference water index (NDWI) to distinguish the characteristics of water and non-water bodies, which can effectively make up for the shortcomings of HSI and VANUI. They then combined the enhanced vegetation index (EVI) and NTL to create NUACI in order to reduce the saturation of night lights.

The above method is based on Landsat and MODIS products to fuse nighttime light data from a natural perspective to extract the built-up areas. The rise of big data has had a significant impact on urban geography; cases in which big data analyses are used are becoming more common [22]. Many scholars apply data that characterize human activities, such as points of interest [23,24], road network data [25,26], and social media data [27,28], to identify urban built-up areas. Point of interest (POI) data are now widely used in urban built-up area extraction in combination with NTL data because of the rich geographical information that they contain. Studies [29] have shown that POI data can compensate for the limitations of NTL data in terms of spatial detail. After intersecting NTL and POI data, Zheng et al. [30] obtained better built-up area accuracy after morphological corrosion and expansion operations than when using a single NTL or POI data source. Another type

of road network data that is closely related to human economic activities is also used in NTL desaturation. Zheng et al. [31] first proposed incorporating road network data into NTL desaturation in 2018. Li et al. [29] used the characteristics of the positive correlation between road network data and night light data, integrated road network data into night light data, and used the threshold method to extract the boundary of the Xuzhou built-up area, obtaining accurate results. In addition, population migration data are also used in NTL to identify built-up areas. For example, Zhang et al. [32] used Baidu migration data and POI data to revise NTL and extract the built-up area of Guangzhou.

Using a single night light data source causes inaccurate extraction results in built-up areas. Previous studies lacked the integration of nighttime lighting data from multiple perspectives. In this study, we introduced POI, EVI, and LST data into NTL. NTL data were introduced into the enhanced vegetation index (EVI); the HSI index was improved using enhanced vegetation data; big data POI were introduced to construct the HP index; and the HP index was introduced into the LST to construct the HPL index. In this manner, we identify urban built-up areas in the six districts of Dalian from multiple dimensions. This will provide a reference boundary for rational land use in Dalian's built-up area in the future. Moreover, it will provide important support for the sustainable economic development of Dalian's six districts.

The research focus of this study was as follows: (1) to explore the spatial distribution characteristics of NTL data after integrating EVI, POI, and LST data; (2) to explore whether the accuracy of NTL in identifying built-up areas by combining multi-source data improved when using the support vector machine method; (3) to establish the differences between NTL and HSI, HSI and HP, and HP and HPL in identifying built-up area boundaries.

## 2. Materials and Methods

### 2.1. Study Area

Dalian is located in the southern part of the Liaodong Peninsula, in the mid-latitude region. It has a resident population of approximately 7.45 million according to the seventh national census. The city has 7 districts, 1 county, and 2 county-level cities under its jurisdiction. As of the end of 2020, the city's afforestation area was approximately 1333 hectares. Dalian is surrounded by the sea on three sides, i.e., the Yellow Sea and the Bohai Sea. To the north, there are the Great Black Mountain and the Hengshan hilly land. The area is composed of a mosaic of mountains and sea. The scope of this study is the urban planning area range selected in the Dalian City Master Plan (2001–2020) (revised in 2017), namely, the six districts of Dalian city (Figure 1): Zhongshan District, Xigang District, Shahekou District, Ganjingzi District, Lushun District, and Jinzhou District. The total area is 2521.26 square kilometers. Among them, Zhongshan District, Xigang District, Shahekou District, and Ganjingzi District are the main urban areas of Dalian city.

### 2.2. Data Preparation

The main data sources for this paper are listed in (Table 1) and (Figure 2).

(1) Hu et al. [33] combined NTL, demographic data, and land use data to establish a stepwise regression model to obtain the population distribution in Sichuan and Chongqing, and the results showed that the correlation between NPP-VIIRS data and the population was stronger than that for DMSP-OLS data. Therefore, NPP-VIIRS data were chosen for the nighttime lighting data in this paper. These data were obtained from the U.S. National Geophysical Data Center, which has a higher radiometric sensor resolution [34,35]. As compared with the DMSP-OLS data, metalight information oversaturation is not a problem. The data from May to August were excluded because the summer light in global mid- and high-latitude regions are highly susceptible to stray light, resulting in zero nighttime light

brightness values for the region. The annual NPP-VIIRS data were synthesized using the monthly data for 2020. The calculation formula is as follows:

$$I_x = \sum \begin{pmatrix} n = 12 \\ y = 1 \end{pmatrix} \frac{L_y}{9} \tag{1}$$

where $I_x$ is the synthetic annual data and $L_y$ is the monthly data of the corresponding month (May to August data were removed). However, the NPP-VIIRS data contain numerous noise problems and outliers, so the masked noise reduction process was performed using the official 2015 U.S. denoised VIIRS annual data. Synthetic denoised NPP-VIIRS data for the year 2020 were obtained using study area vector data mask extraction.

(2) The EVI data are derived from MOD13Q1 and have a temporal resolution of 16 days and 22 data periods in a year for annual mean synthesis. The synthesized data results are stable and less affected by extreme anomalous values.

(3) The POI data came from the Gaode map provided in 2020 on the official application. The collection time was December 2020. A total of 240,679 POIs were crawled from a total of 23 POI categories in Dalian city in 2020 after cleaning and checking the data;

(4) LST data were obtained by inversion calculations of Landsat8 data using the Google Cloud Computing Platform (Google Earth Engine, GEE), which is capable of storing and processing huge amounts of data [36]. The time range was from 1 January 2020 to 31 December 2020, and the image data with clouds were removed. The final data were obtained by averaging. Mask extraction was performed using the administrative boundary of the study area.

(5) The 0.6 m resolution Google HD remote sensing image was acquired in January 2021. All of the above data were projected to WGS-UTM-1984 with a uniform spatial resolution of 30 m.

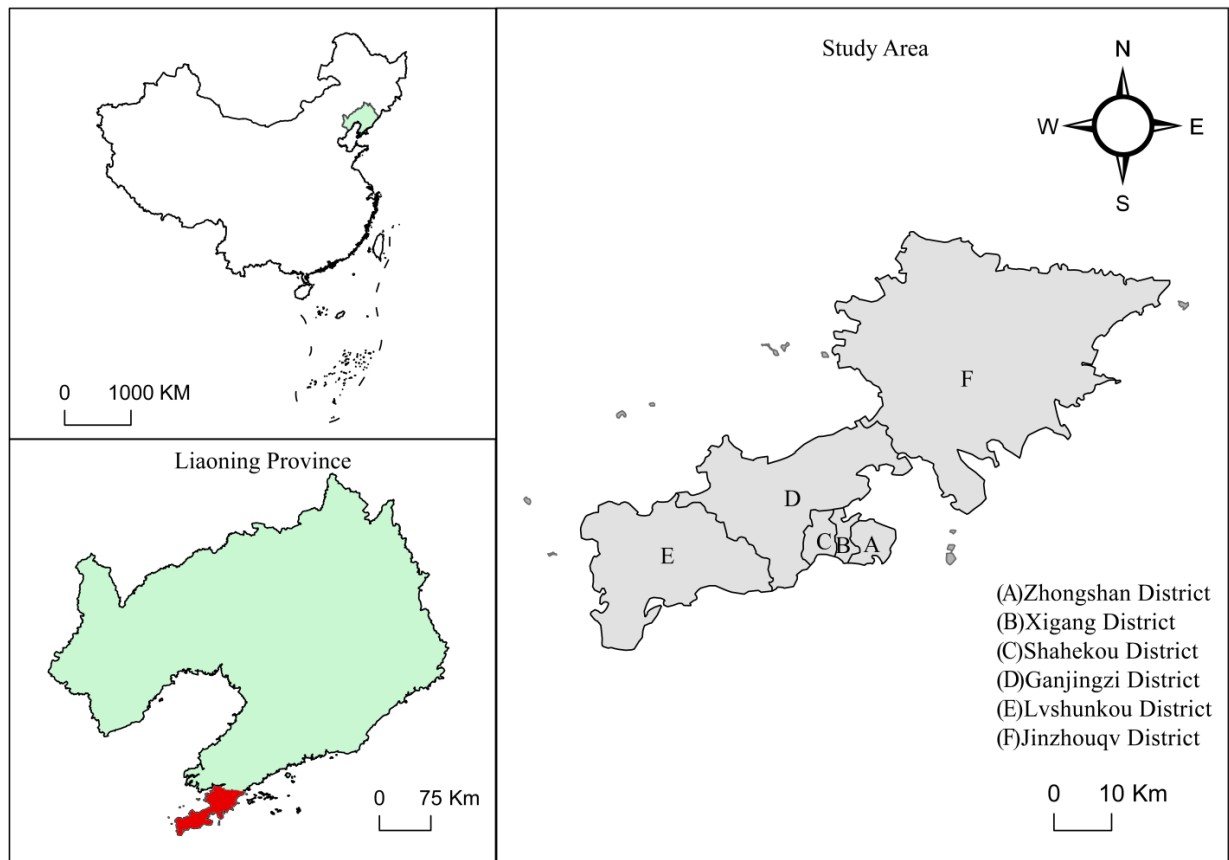

**Figure 1.** Overview of the study area.

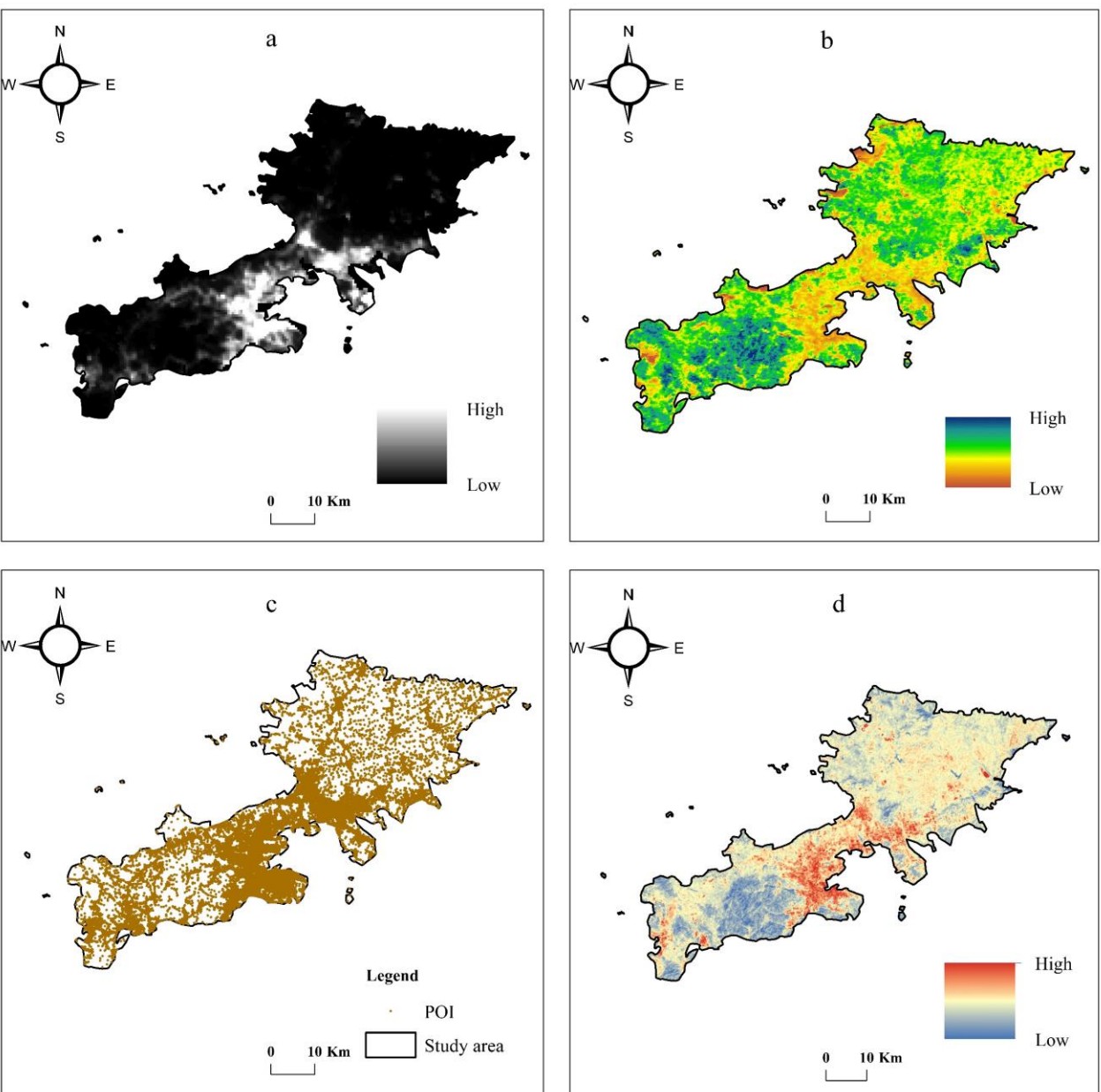

**Figure 2.** Experimental data: (**a**) NPP-VIIRS nighttime lighting data; (**b**) EVI data; (**c**) POI number; (**d**) LST data.

## 2.3. Method

### 2.3.1. POI Kernel Density Estimation Bandwidth Threshold

The kernel density estimation method estimates the density of function point elements in the surrounding neighborhood using the kernel density, which is a method for high-quality density estimation [37].

$$P_i = \frac{1}{n\pi h^2} \sum_{j=1}^{n} W_j \left( 1 - \frac{D^2_{ij}}{h^2} \right)^2 \tag{2}$$

where $P_i$ is the estimated density value of estimation point I; $h$ is the width of the regular neighborhood around the point of study object $j$, i.e., the bandwidth; $n$ is the number of sample points in the bandwidth; $D_{ij}$ is the distance between the point to be estimated and the study object $j$; $W_j$ is the weight of the study object.

**Table 1.** Shows the data for the categories mentioned above.

| Data | Resolution | Source | Acquisition Date |
|---|---|---|---|
| NPP-VIIRS | 500 m | http:/ngdc.noaa.gov/eog/ | 9 months in 2020 |
| EVI | 250 m | https://ladsweb.modaps.eosdis.nasa.gov | 12 months in 2020 |
| POI | | https://lbs.amap.com/ | 31 December 2020 |
| LST | 30 m | https://code.earthengine.google.com/ | 12 months in 2020 |
| Google Maps | 0.6 m | www.Amap.com | 1 January 2021 |

### 2.3.2. HSI, HSI and POI, HSI and POI, and LST Index Construction

The identification of urban built-up areas is only limited to a single perspective, which will inevitably cause the identification results to be unobjective [32,38]. Therefore, in this study, we constructed HSI, HSI and POI, and HSI and POI and LST indices for the comparative analysis of urban built-up areas.

(1) HSI index construction. HSI is the proportion of human settlements per unit image element. This can characterize the suitable space for the existing population to live in and is used to reflect the density of the population. The NPP-VIIRS data represent the intensity of regional lights at night: the brighter the lights are [39], the more concentrated the population is in the distribution area. However, NDVI data are prone to saturation, especially in areas with high vegetation cover, which makes it difficult for NDVI to distinguish the background soil noise. EVI addresses this problem by using the soil-adjusted vegetation index to reduce the influence of the soil background and improve the error caused by atmospheric or vegetation growth, which is less likely to occur than when using NDVI [40]. Therefore, the HSI formula was improved. After the nighttime light data are normalized, the formula is as follows:

$$NTL_{nor} = \frac{NTL - NTL_{min}}{NTL_{max} - NTL_{min}} \tag{3}$$

where $NTL_{nor}$ is the normalized night light value, $NTL_{max}$ is the maximum nighttime light index, and $NTL_{min}$ is the minimum nighttime light index.

The *HSI* calculation formula is as follows:

$$HSI = \frac{(1 - EVI) + NTL_{nor}}{(1 - NTL_{nor}) + EVI + NTL_{nor} * EVI} \tag{4}$$

(2) HP index construction. The spatial resolution of NPP-VIIRS and the *EVI* raster data is low. Therefore, in this study, we introduced vector POI data to integrate the *HSI* index. This improved the low raster data resolution and the extraction accuracy of urban built-up areas as it incorporated the accurate location information and attribute POI information. The POI data itself exhibit a certain stability and validity. The built-up area extraction accuracy can be further improved by removing the noise from lights at night [41]. Since there is a large difference in magnitude between the HSI index and the POI kernel density, the above two types of data need to be normalized to unify the data of different magnitudes under one order of magnitude. This makes it possible to take advantage of various types of data and is necessary to avoid large errors in the experimental results. The normalization formula is as follows:

$$X' = \frac{X - min(X)}{max(X) - min(X)} \tag{5}$$

where $X'$ is the normalized value, $min(X)$ is the minimum value, and $max(X)$ is the maximum value.

The geometric mean was mostly used for image fusion in previous studies because it can effectively eliminate the effect brought about by image extremes in image fusion. In addition, it makes it possible to retain the original image information to a decent degree [42,43]. Therefore, the geometric mean method was used to construct the *HP*

synthesis. The *HP* composite index and HPL index were constructed using the geometric mean method. The calculation formula is as follows:

$$HP = \sqrt[2]{HSI \times POI} \tag{6}$$

where *HP* is the normalized *HP* value at the point I, and P is the normalized kernel density value of *POI* at point i.

(3) HPL index construction. *HSI* characterizes the actual distribution information of the population in a built-up area, *POI* reflects the socioeconomic factors and municipal utilities of the city, and *LST* reflects the increase in surface temperature due to the continuous development of built-up areas in the city. Therefore, *LST* was introduced into the construction of the HSI_POI index.

$$HPL = \sqrt[3]{HSI \times POI \times LST} \tag{7}$$

where *HSI* is the normalized *HPL* value at point i, $P_i$ is the normalized kernel density value of *POI* at point i, and *LST* is the normalized *LST* value at point i.

### 2.3.3. SVM-Based Supervised Classification to Extract Built-Up Areas

Support vector machine (SVM) is a method to determine the class of unknown image elements using a field survey or visual interpretation [44]. SVM is a powerful machine learning algorithm that is stable, easy to use, and minimizes structural risk. After radiometric calibration and atmospheric correction of Landsat8 data in ENVI, a region of interest is created. Four site types were determined using the support vector machine method: built-up areas, bare land, water bodies, and vegetation. All had a sample separation of 1.86 or higher. Reclassification was performed in ArcGIS to classify all bare soil, vegetation, and water bodies as non-built-up areas.

### 2.4. Accuracy Evaluation

Because the extent of urban built-up areas is not represented by a fixed boundary, it is also impossible to accurately calculate this in the field [32]. Therefore, the study used the SVM-supervised classification method to identify urban built-up areas using visual interpretation of Landsat 8 remote sensing image data. A total of 1000 sample points were randomly generated within the six districts to avoid randomness in the obtained data. The distance between each random point was always greater than 5 m. Using 0.6 m resolution Google remote sensing images once more for the visual judgment, 250 points were generated within the urban physical area and 750 points were generated within the non-urban area (Figure 3). The layers were classified to create points of interest and a confusion matrix was built to evaluate the experimental results. In this paper, two metrics, i.e., overall accuracy (OA) and kappa coefficient, were used to evaluate the accuracy of built-up areas. The calculation formula is as follows:

$$P_o = \frac{T_r}{N} \tag{8}$$

where $P_o$ is the overall classification accuracy, $T_r$ is the number of correctly classified pixels, and $N$ is the total number of pixels in the study area.

$$P_e = \frac{a_1 \times b_1 + a_2 \times b_2 + \ldots + a_c \times b_c}{N} \times N \tag{9}$$

$$K = (P_o - P_e)/1 - P_e \tag{10}$$

where *K* denotes the kappa coefficient, $P_o$ represents the overall classification accuracy, and $P_e$ is denotes the expected consistency when two annotators are randomly assigned labels.

In this study, EVI, POI, NTL, and LST are fused to construct NTL, HSI, HP, and HPL indexes. The urban built-up areas were extracted by SVM and the built-up areas of each index were extracted for comparative analysis. The experimental flow is shown in Figure 4.

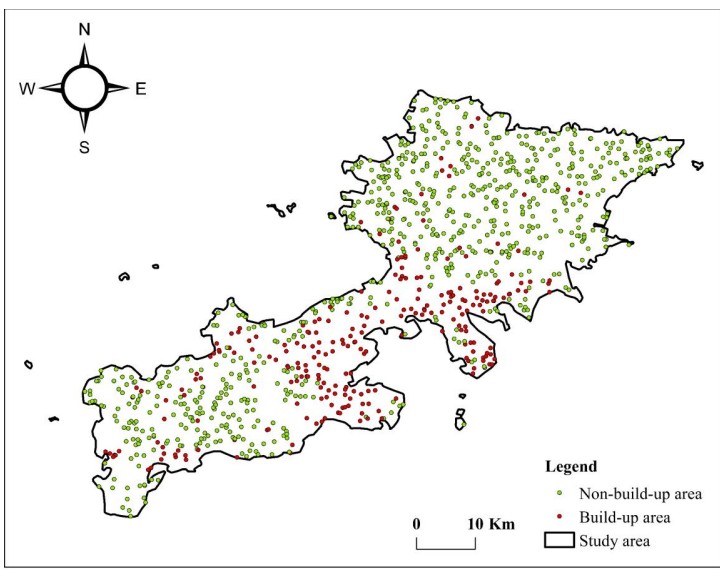

**Figure 3.** Random point validation in the study area.

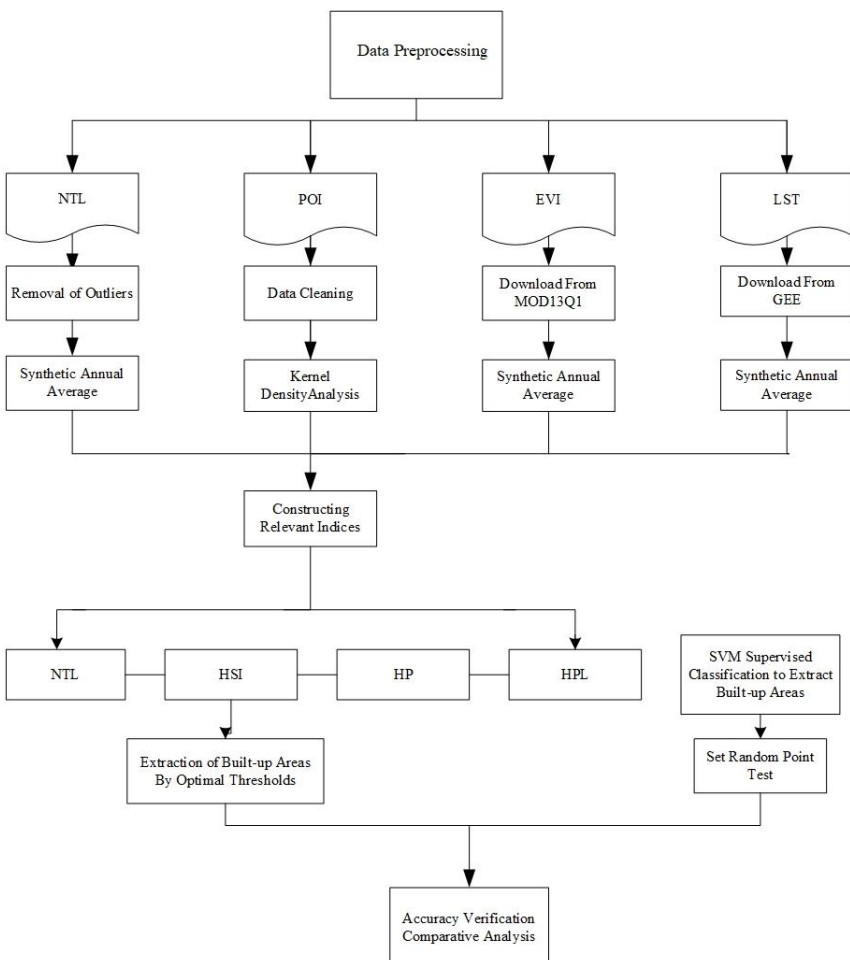

**Figure 4.** The Flow Diagram of the Study.

## 3. Results

The combination of multi-source remote sensing data and NTL data can effectively mitigate the blooming effect of NTL. EVI and LST data can show the decreasing vegetation and surface temperature from the center of an urban built-up area outward from a natural perspective. The density of POI, on the other hand, shows a continuous decrease from the center of the urban built-up area outward from an economic perspective. By integrating EVI, LST, and POI data into NTL and constructing an HPL index with natural, economic, and demographic attributes, the nighttime lighting data can be corrected from both natural and economic perspectives.

### 3.1. Analysis of POI Kernel Density Results

The choice of POI kernel density bandwidth will affect the results in different ways [37]. Thus, referring to previous studies [45], in this paper, three nucleus density bandwidths were selected for comparison (Figure 5). When the 1000 m bandwidth was used, the spatial distribution of the POI nucleation density was scattered and there were more internal voids, which is not conducive to the overall analysis of nucleation density. When the 1500 m bandwidth of nucleation density was used, the number of holes in the lower nucleation density area was greatly reduced. When the 2000 m bandwidth of nucleation density was used, the POI nucleation density map space was too smooth and the actual situation was not reflected at the edges. Therefore, we chose the 1500 m bandwidth for this study.

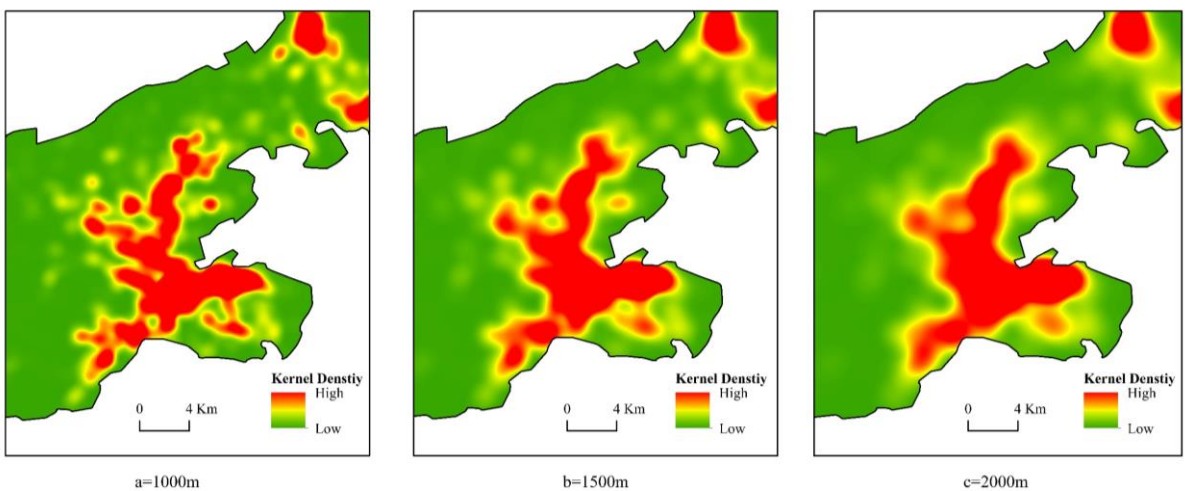

**Figure 5.** Kernel density estimation results for different bandwidth cases.

### 3.2. Spatial Distribution Characteristics of NPP-VIIRS, HSI, HP, and HPL

Figure 6 shows the spatial distribution of the NPP-VIIRS, HSI, HP, and HPL indices. The red indicates the high-value area, which reflects the high population density of a certain area, the close connection of economic activities, and the predominant impervious ground cover surface. The green indicates the low-value area, which mainly represents areas with a low population density, more vegetation cover, less connected economic activities, and mostly non-impermeable surfaces.

The high-value areas are mainly concentrated in the west of Zhongshan Square in Zhongshan District, the middle of Xigang District, and the east of Shahekou District. In addition, high-value areas also appear in the vicinity of large transportation hubs, including Ganjingzi District, Dalian Zhoushuizi International Airport, Dalian North Station, Dalian Station, and other transportation hubs, such as Jinzhou District. High index values are mainly distributed in the northern and southern parts of Jinzhou city, with the southern part of the city containing the Dalian University of Arts, Dalian University of Nationalities, and several shopping centers. The higher values in the Lushun district are near the Lushunkou bus station. There are also some small-scale high-value areas distributed

away from the urban area. Using Google HD images for comparison, we found that these included the Xiaoyaowan International Business District in Jinzhou District, the Tieshan New Town area in the west of Lushun, and the Shuishiying New Town area in the north. These findings are consistent with the key development directions defined in the Dalian City Master Plan (2001–2020) (revised in 2017). This indicates that great improvements have been seen in relation to the population density, infrastructure construction, and urban development in these areas as compared with the surrounding areas. These high-value areas are distributed in the aforementioned areas, mostly because the area has several shopping districts with a wide range of shops and historical precipitation. These include Xi'an Road shopping district, Xiangxu Reef shopping district, and China Road shopping district. These shopping districts contain many large shopping centers, and these drive the economic development of the surrounding areas. Zhongshan District, Xigang District, and Shahekou District are the most densely populated areas in Dalian, and according to the data from the Seventh National Census in 2020, the population density of Zhongshan District was 9184 people/square kilometer, that of Xigang District was 11,643 people/square kilometer, and that of Shahekou District was 17,876 people/square kilometer. The population is densely distributed and there are railway stations, bus stations, and airports in the main urban area. These areas contain a dense road network, well-developed infrastructure, and convenient travel conditions. In addition to attracting residents to spend money, a large number of foreign tourists come to shop every year.

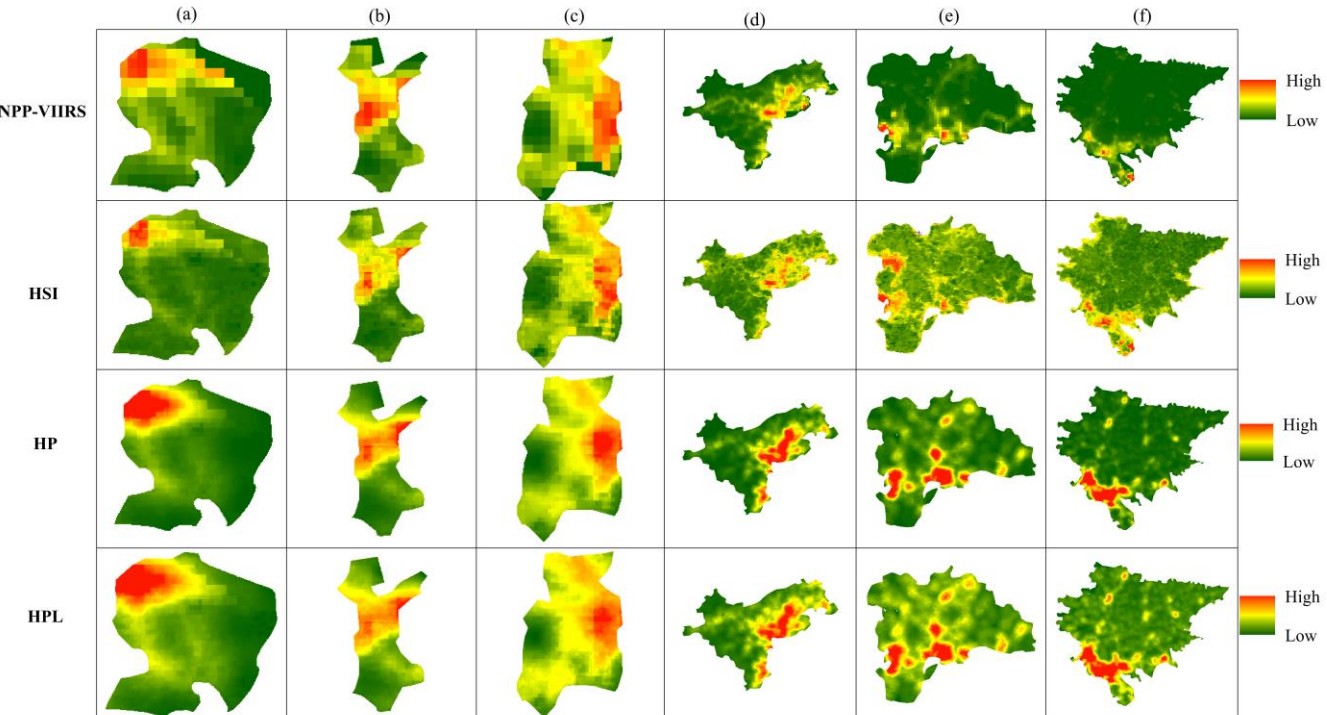

**Figure 6.** Spatial distribution images of NPP-VIIRS, HSI, HP, and HPL in selected areas of the six districts in Dalian: (**a**) Zhongshan District; (**b**) Xigang District; (**c**) Shahekou District; (**d**) Ganjingzi District; (**e**) Lushunkou District; (**f**) Jinzhou District.

However, the spatial distributions provided by NPP-VIIRS, HSI, HP, and HPL in the six districts of Dalian city were slightly different, and the NPP-VIIRS images exhibited an obvious "spillover effect", especially in Zhongshan District, Xigang District, and Shahekou District, where the images appeared blurred. This was mainly due to the low spatial resolution of NPP-VIIRS images. For example, the spatial texture details of HSI images are significantly better than those of NPP-VIIRS images, which is mainly due to the adjustment of NTL using the EVI index. HPL images inherit the advantages of HSI and HP images,

and the introduction of LST data made the images richer in spatial details as compared with the HP images. In addition, it enhanced the local road information identified by HP, such as in Lushunkou and Jinzhou districts.

### 3.3. Visual Interpretation to Extract Urban Built-Up Areas

NPP-VIIRS, HSI, HP, and HPL indices were compared with the Landsat8 classification results several times, and multiple thresholds were artificially set with 1000 sampling points to calculate the confusion matrix. Table 2 shows the OA and kappa values of the extracted urban built-up area boundaries under different data sources. NPP-VIIRS achieved the optimal overall classification accuracy of 88.81% and a kappa coefficient of 0.67 when the threshold value was 9. When the threshold value was greater than 9, the OA and kappa coefficients exhibited a slowly decreasing trend. The overall accuracy and the kappa values did not improve as compared with NPP-VIIRS, because HSI tended to mix large areas of bare soil into urban built-up areas, causing a decrease in accuracy when identifying built-up areas. After incorporating POI data into HSI, the overall accuracy and kappa coefficient reached the maximum value of 90.01% and 0.72, respectively, when the threshold value was 0.02, while the OA and kappa decreased as the threshold value increased above 0.02. The best overall accuracy and kappa coefficient for the HPL index with fused multi-source data were 91.41% and 0.77, respectively, when the threshold value was 0.05. In addition, the overall classification accuracy was maintained above 80% with the increase in the threshold value. The kappa coefficient decreased slowly as compared with the HP index.

**Table 2.** HSI, HP, and HPL indices' OA and kappa values.

| Index | Optimal Threshold | Overall Accuracy | Kappa |
|:---:|:---:|:---:|:---:|
| NPP-VIIRS | 9 | 88.81% | 0.67 |
| HSI | 0.85 | 87.21% | 0.64 |
| HP | 0.02 | 90.01% | 0.72 |
| HPL | 0.05 | 91.41% | 0.77 |

### 3.4. Comparative Analysis of Constructed Area Boundary Extraction

The NPP-VIIRS, HSI, HP, and HSI thresholds, when the best overall classification accuracy and kappa coefficients were obtained, were used to generate the urban built-up area boundaries. In addition, NPP-VIIRS with HSI, HSI with HP, and HP with HPL were analyzed for comparison. Table 2 shows the OA and kappa values under the optimal thresholds of the NPP-VIIRS, HSI, HP, and HPL indices.

#### 3.4.1. Comparative Analysis Based on the Results of the NPP-VIIRS and HSI Index

The boundaries extracted using NPP-VIIRS with a threshold of 9 were compared with those extracted using HSI with a threshold of 0.85. Figure 7 shows the built-up areas identified by NPP-VIIRS. It has a distinct jagged shape and only small built-up areas were identified far from the patches of built-up areas. In contrast, the built-up area patches identified using the HSI index are fragmented and fine-grained and exhibit a lack of coherence as compared with the NPP-VIIRS image. However, the edge complexity of the HSI built-up area increased, demonstrating a richer edge detail. For example, in the southeastern part of Zhongshan District, HSI better identified built-up areas and vegetation as compared with NPP-VIIRS. If there was a lot of bare soil in the urban area, the HSI index method should be avoided as it can mistake bare soil areas for built-up areas.

#### 3.4.2. Comparative Analysis Based on the Results of the HSI and HP Index

The extracted boundary with an HSI threshold of 0.85 was compared with the extracted boundary with an HP threshold of 0.02. The HSI index extracted urban built-up areas with broken internal details and many holes of different sizes. These do not match the definition of built-up areas mentioned in the previous section. The HP index method addressed these problems well, and it was able to extract built-up areas in a piecewise distribution with a

good integrity. The difference between the urban built-up areas extracted using the HSI index and the results extracted using the HP index was large (Figure 8), and after a Google map comparison, it was found that most of the differences were in coastal areas, such as the northwest and north of Ganjingzi District in the main urban area, the west and north of Jinzhou District, and the west of Lushunkou District, i.e., mostly mudflats, seawater farms, salt farms, bare soil, and Dalian Dayaowan Free Trade Zone. These areas are characterized as having a high light intensity at night and a lack of vegetation cover on the ground cover. Thus, the HSI index identified them as urban built-up areas. The enhanced vegetation index had difficulty distinguishing bare soil from urban built-up areas. This is why the accuracy of the built-up area boundary extracted using HSI was lower than that of NPP-VIIRS.

### 3.4.3. Comparative Analysis Based on the Results of the HP Index and HPL Index

The boundaries extracted at an HP threshold of 0.02 were compared with the boundaries extracted at an HPL threshold of 0.05. Some differences exist for local areas as shown in Figure 9. There were three main categories of areas. Taking class A, an industrial park, as the representative area, the built-up area identified using the HP index was extremely small. After the comparison with Google Maps, it was found that the HPL index was able to identify a range of surface buildings. There are more than ten factories and five residential communities distributed throughout the industrial park in area A, with some supporting convenient facilities. In addition, in the western part of Yingchengzi Street in Ganjingzi District, the built-up area identified using the HPL index was cluster-shaped, and the HPL index did not identify the area. In the field investigation, it was found that this area is a hub for educational institutions, such as Dalian Ocean University and Liaoning Police College. However, the number of POIs in the above areas is low. Industrial parks, residential areas, and university gathering places are covered with hardened roads with buildings, so this area should have been identified as an urban built-up area.

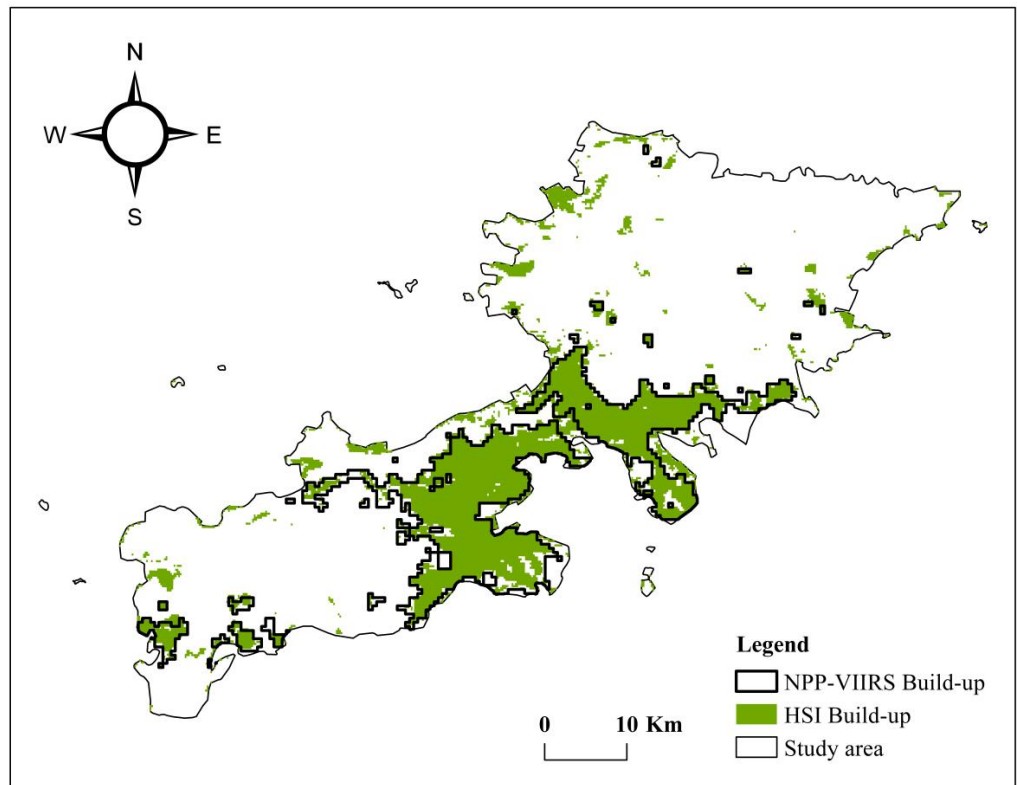

**Figure 7.** Comparison of the built-up area boundaries identified using NPP-VIIRS and HSI.

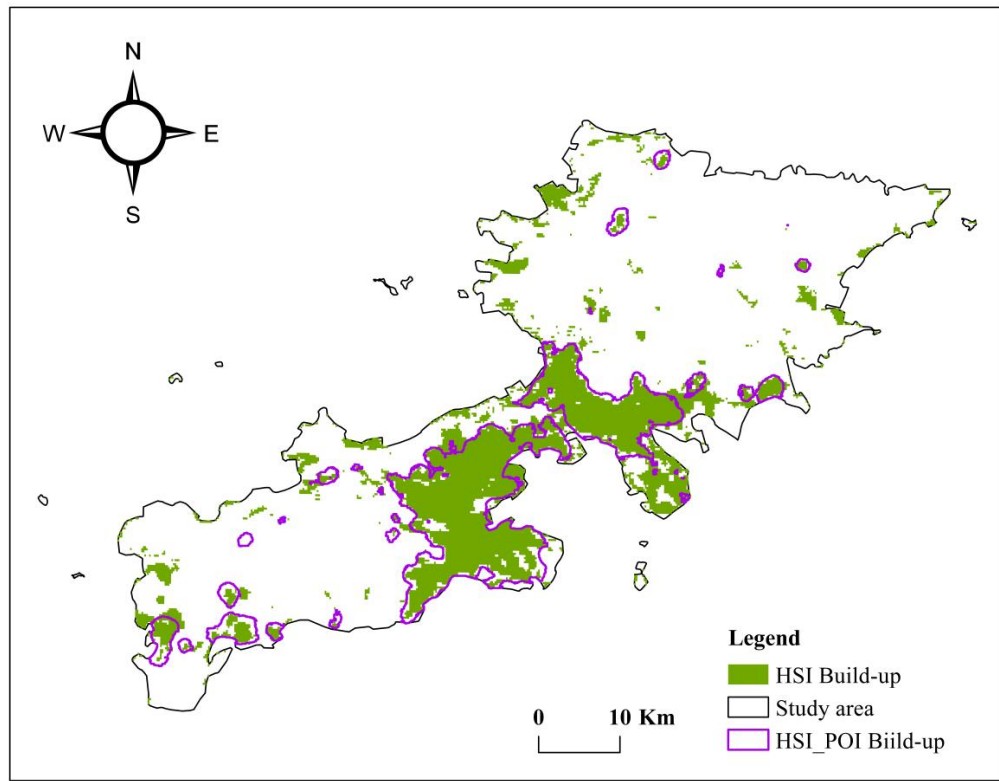

**Figure 8.** Comparison of the urban built-up area boundaries extracted using the HSI and HP indices.

Area B is the Dayaowan Free Trade Zone, which mostly contains petrochemical companies and a single type of POI. Thus, the nuclear density value of this site was low compared to other areas and the HP index did not identify it as an urban built-up area. The same situation was also found in area C, which comprises several petrochemical companies, low surface vegetation coverage, and many hardened roads, with large plants and open-air chemical equipment in the area. The area is in the vicinity of a built-up area and is closely connected to the surrounding built-up areas, so it was also identified as an urban built-up area.

The areas D and E are situated at the junction between urban and rural areas. After the field survey, it was found that area D is mostly made up of a logistics company, and area E is mostly an industrial park, with relatively complete transportation and plant facilities. There are several building communities in the area, although the number of POI types decreases compared to the number in the inner city. In area E, located in the northwest of Ganjingzi, the range area identified by the HPL index is larger than the HP index, and the number of POI types was significantly reduced compared to the urban center. This area contains several residential communities and the roads inside are dense in order to meet the daily travel needs of the residents. The high HSI values in both areas indicated a high population density, and the surface temperature data were significantly higher than those of the surrounding areas, so these areas were also classified as urban built-up areas.

After the analysis, it was found that the urban built-up area edge details identified using the HPL index were accurately portrayed, basically showing the urban entity edge details, while the boundary of the urban built-up area identified using the HP index was too smooth. For certain built-up areas far from a city center where there are many industrial parks, bonded areas, and urban–rural areas, the HPL index should be used.

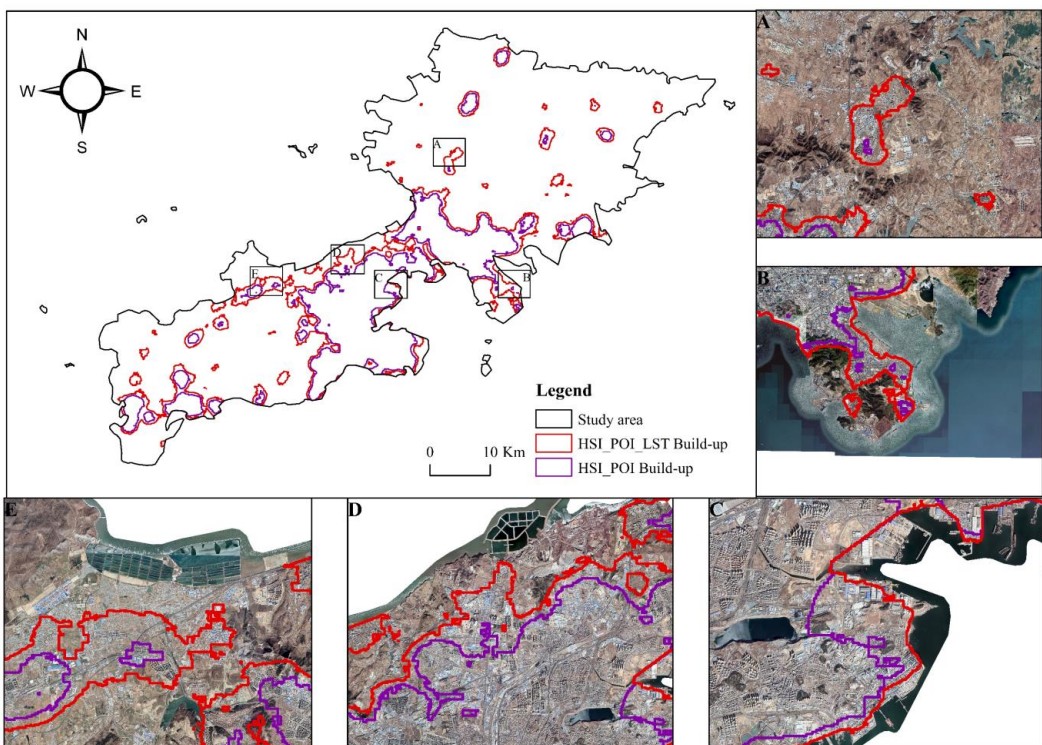

**Figure 9.** Comparison of the HP index and HPL index identifying the boundary of the built-up area. Area (**A**) an industrial park. Area (**B**) Dayaowan Free Trade Zone. Area (**C**) comprises several petrochemical companies. Area (**D,E**): situated at the junction between urban and rural areas. Area (**D**) logistics company. Area (**E**) an industrial park.

## 4. Discussion

### 4.1. HPL Index for Urban Built-Up Area Extraction Advantage

The extraction of urban built-up areas is an important method with which to detect urban changes. It can be used to keep track of the dynamics of the development of built-up areas and is important for urban problem-solving and urban planning and construction. Urban built-up areas are complex systems and their boundaries are constantly changing in dynamic expansion [45]. At this stage, the extraction of urban built-up areas is mostly based on the natural condition of ground cover or socio-economic elements, such as Y. zha et al. [46] who extracted the built-up areas of Nanjing city from LandsatTM images using a normalized building index and a normalized vegetation index based on the ground cover condition; but, there is a problem in mixing the built-up areas and non-built-up areas (vegetation and water bodies). Another perspective is socioeconomic, using demographic data and social media data. Zhou et al. [47] proposed to specify the population density of 2000 people/square kilometer as the territorial boundary of urban entities; the update speed of large-scale demographic data is slow. Along with the development of big data nowadays, social media data are also applied to built-up area identification; but, its main user group is limited to young and middle-aged people, and there is a lack of data on the elderly population and children.

Nighttime lighting data can capture human activity activities in real time and they have been widely used to extract urban built-up areas. However, nighttime lighting data are characterized as having a low spatial resolution. Thus, researchers have introduced EVI into NTL to address this problem. EVI can effectively reduce the interference related to the soil background as compared with NDVI [48]. However, since both bare land and urban built-up areas are not covered by vegetation, errors are possible when determining the boundaries of built-up areas using only the EVI index to reduce the saturation of NTL. With the continuous development of urban areas, there is a large amount of impermeable surface cover inside the city and the urban heat island effect caused by the impermeable

surface is positively correlated with LST [49]. LST is closely related to land cover and land use and can be used to further distinguish between urban and rural areas [50,51]. Various studies used NDVI, NTL, and LST data to examine the spatial and temporal aspects of several urban built-up areas on a large scale [52,53]. However, these methods cannot be applied to the extraction of urban built-up areas on small and medium scales. With the rise of big data, certain studies introduced POI data or road network data into NTL to identify urban built-up areas [54,55]. However, such studies only focus on the influence of socioeconomic factors on urban development.

This study is innovative in that we constructed an HPL index, which integrates natural and social elements such as LST, HSI, POI, etc. The HPL index can comprehensively identify urban built-up areas from both natural and social perspectives. In addition, it was found that the HPL index provides richer spatial details of built-up areas as compared with single or double data sources and improves the spatial resolution of a single night light data source. Our method produced the highest OA accuracy and kappa coefficient among the four methods in terms of identifying urban boundaries. The connectivity of the identified built-up areas was shown and the edges of the built-up areas contained rich detailed information.

### 4.2. Selection of Support Vector Machine Methods

The most common extraction methods for nighttime lighting data are threshold methods, such as the empirical threshold method, the mutation detection method, the statistical data comparison method, etc. Although these methods are simple and easy to operate, the threshold extraction results are often larger than the actual built-up area. The current popular deep learning method can also be applied to the extraction of built-up areas, but this method suffers from the typical "black box problem" [56] and requires tens of thousands of samples to extract reliable results. Unlike the above methods, the SVM method is based on Landsat remote sensing images with rich spectral information and uses a priori knowledge to select a small number of representative samples. Through machine learning algorithms, it can quickly obtain urban and non-urban areas and it is easy to operate.

### 4.3. Constraints

This study has several shortcomings: (1) The resolution of the NPP-VIIRS data selected in this paper is low. In the future, we hope to combine these data with higher resolution nighttime lighting data, such as the nighttime remote sensing lighting data from Luojia. (2) Our study is a static study. The dynamic development and expansion of urban built-up areas will be studied in the future using dynamic data such as population migration data.

### 5. Conclusions

Taking Dalian city as an example, in this study, we used NPP-VIIRS, EVI, POI, and LST data to extract built-up areas using kernel density analysis, image geometric mean, and support vector machine-supervised classification recognition methods. A total of 1000 sampling points were created in Google HD remote sensing images for verification. The NPP-VIIRS, HSI, HP, and HPL indices were compared and analyzed. The HPL index constructed in this study improves the nighttime light spillover effect compared with a single data source or two data sources, and the OA and kappa accuracy of this index is better than that of the NPP-VIIRS, HSI, and HP indices, and also obtains rich edge information. The HPL index constructed in this paper provides new ideas for urban planners to study the city of Dalian, and the HPL index can be used to study the dynamic changes of the built-up area of Dalian in a long term series in the future, and if we want to obtain richer spatial details inside the built-up area, we can consider combining with Luojia 1-01 nighttime light imagery. This is of great significance to the healthy development, planning, and reconstruction of urban areas in the future.

**Author Contributions:** Conceptualization, X.L.; software, Y.S. and H.L.; validation, X.H.; writing—original draft, Y.S.; writing—review and editing, X.L. and Y.S.; visualization, Y.S.; supervision, X.L. All authors have read and agreed to the published version of the manuscript.

**Funding:** This research was funded by the National Natural Science Foundation of China (grant number 41671158).

**Data Availability Statement:** The data presented in this study are available on request from the corresponding author.

**Conflicts of Interest:** The authors declare no conflict of interest.

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
