# Peer review of "Extraction of Urban Built-Up Areas Using Nighttime Light (NTL) and Multi-Source Data: A Case Study in Dalian City, China"

_land, doi:10.3390/land12020495_

Round 1

Reviewer 1 Report

Dear Authors,

the text of the article is incomprehensible in many places.

I suggests submitting the article to MDPI's Language Correction.

A few comments:

Section – Abstract: lines 30-33. Some parts of the text require editorial editing. For example:

…. HPL has population, interest point

Line 37: Since the reform and opening - clarify what this means.

Lines 40, 41:  errors in references to literature.

Line 65:DN - not use abbreviations - solve them earlier.

Line 72: some scholars have tried to introduce their remote sensing image products ???? Who is "scholars" ? What kind of RS products did they have ?

Line 77: However, the number of high resolution images started late … What does it mean ? Is it about the first VHRS systems and the availability of the images they acquire ?

Author Response

Thank you for your suggestion. This paper has been professionally English edited by MDPI English Editing service and is highlighted in the text. It would be appreciated if you could help check and provide us with some comments.

Reviewer 2 Report

The paper has focused on a very interesting topic 'the extraction of urban build-up areas' and proposed a hibrid index using NTL data and POIs data, to improve the accuracy of the extraction process. The topic is well related with the journal and the experiment is well designed. However, there are still several major issues need to be fixed before the paper can be further published.

1. Why is it so important to extract the urban build-up area? I think you can improve the current expression in first graph and provide more evidence(related to SDGs can be considered).

2. I recomment the authors to better construct the Introduction part. You can conclude the previous methods into several catogories and make some comparision. Moreover, I think it might be interesting to introduce more about how previous researches adope POIs/social media/human activities data in this topic(I have read some before).

3. The references style seems to be wrong in the whole paper! Please check it carefully (page 2 Line 41 for instance).

4. In Method section, you can put a flowchart to better document your experiment.

5. Page 13.It's great that you made comparision among VIIRS, HSI and HP indexes besides your own proposed index. But I think you can provide more details(or figures) to help us understand which method(or index) we should adopt in specific situation.

6. In conclusion part, the conclusions(point 1-3) are too long. You can generate some key findings in this part, rather than put too much experimental results.

7. Please improve your english expression and revised some typos carefully. 

Author Response

Thank you for your letter and for the reviewers’ comments concerning our manuscript entitled “Extraction of urban built-up areas based on nighttime lighting data with improved multi-source data” (land-2166783). Those comments are all valuable and very helpful for revising and improving our paper, as well as the important guiding significance to our researches. We have studied comments carefully and have made correction which we hope meet with approval. Revised portion are marked in blue in the paper. The main corrections in the paper and the responds to the reviewer’s comments are as flowing:

  • Point 1: Why is it so important to extract the urban build-up area? I think you can improve the current expression in first graph and provide more evidence(related to SDGs can be considered)..

Response 1: We have re-written this part according to the Reviewer’s suggestion. (line 29-34, in blue)

  • Point 2: I recomment the authors to better construct the Introduction part. You can conclude the previous methods into several catogories and make some comparision. Moreover, I think it might be interesting to introduce more about how previous researches adope POIs/social media/human activities data in this topic(I have read some before).

Response 2: We have re-written this part according to the Reviewer’s suggestion. (line 81-94,line 98-100,line 106-115 in blue).

In the introduction section, we have added many cases, and the overall idea is to firstly summarize the single night light data to extract the built-up area, secondly to combine the night light data with remote sensing data to extract the built-up area from the natural perspective, and thirdly to combine the night light data with social media, points of interest and other big data to extract the built-up area from the economic perspective.

  • Point 3:The references style seems to be wrong in the whole paper! Please check it carefully (page 2 Line 41 for instance).

Response 3: According to the suggestions of reviewers.We have changed the format of all references

  • Point 4:In Method section, you can put a flowchart to better document your experiment.

Response 4: Considering the Reviewer’s suggestion, We put a flowchart. (line 294 ).

  • Point 5: Page 13.It's great that you made comparision among VIIRS, HSI and HP indexes besides your own proposed index. But I think you can provide more details(or figures) to help us understand which method(or index) we should adopt in specific situation.

Response 5: Considering the Reviewer’s suggestion, We added content related to the study topics / content in the introduction. (line 411-412, and line 476-479 in blue).

  • Point 6: The conclusion part, the conclusion (points 1-3) is too long. Instead of placing too many experimental results, you can generate some key findings in this section.
    Response 6: Considering the reviewer's suggestion, we have deleted the conclusion part of the paper according to your suggestion. (Lines 558--582, blue).

    Point 7: Please improve your English expression and carefully correct some typos.
    Response 7: Thanks for your suggestion. This article has been professionally embellished by MDPI's retouching agency and is highlighted in the text

    We are very fortunate to invite you to review our article as a reviewer and thank you for your informative and valuable comments. Special thanks for your good review. We tried our best to improve the manuscript and made some revisions to the manuscript. These changes do not affect the content and framework of the paper.

Reviewer 3 Report

I read the manuscript carefully I believe the paper could be further strengthened by changes which should be made before it gets published as follows:  

The title

1.      The title should attract the reader and concisely reflects the actual content of the authors’ contributions. However, I found the title needs some modification particularly the phrases “based on…..with”. the title should be changed into (Extraction of urban built-up areas using nighttime light (NTL) and multi-sources data) or similar.

2.      Similarly, I suggest changing the title to include the case study by inserting the name of the geographical area (Dalian city, China): [Extraction of urban built-up areas using nighttime light (NTL) and multi-sources data: A case study in Dalian city, China]

The Abstract

1.      I think several keys are missing from the abstract mainly the motivations of doing this research, gaps to bridge, policy implications of the paper, and contribution to knowledge.

2.       The methodology and data analysis are not clearly represented in the abstract. There is a big difference between data collecting and methodology adoption. Accordingly, I suggest rewriting the abstract consciously and insert a couple of sentences to mention these issues.

Introduction

1.      The introduction section involves the theoretical framework. However, it has not been written in cohesion nor a logical flow style.

2.      The citations and referencing (line 40 for example) is NOT written in a suitable manner. Please correct these mistakes and typo errors. 

3.      The author should mention how the previous studies represented and reviewed the concept of using nighttime light to extract built-up areas within China and elsewhere.

4.      Line 79-80 should be rewritten as the sentence starts with “And”.

5.      Similarly, Lines 81-82 represent a run on sentence [The Normalized Difference Vegetation Index (NDVI), normalized 81 difference built-up index (NDBI), Normalized Urban Areas Composite Index 82 (NUACI)12, and Land Surface Temperature (LST)].

6.      The authors need to enrich the literature review by including more case studies particularly emphasizing the introduced methods and techniques to correct the problems of nighttime light saturation and blooming.

7.      The authors should develop a paragraph which summaries surveying the literature and what this study contributes to the literature.

  Methods

1.      In Figure 1, the authors should represent the study area relevant to the entire country.

2.      The caption of Table1should be changed.  

3.      The results of kernel density estimation should be moved to the finding section.

Findings

1.      The results of the confusion matrix should be represented in a table better than figures (figure 6) and thus table 2 should be moved to section 3.2.

2.      How the authors did correct the NTL dataset?

3.      The resolution and creations of the figures are both very poor.

Discussion

1.      Typo mistakes in line 469 (Nighttime lighting data can capture human activity activities in real-…..).

2.      The authors should discuss the limitations of other datasets not only the NTL.

Conclusions

In the last section of the paper, authors should include recommendations for additional research in the area and I suggest referring to the importance of conducting similar studies which take into account additional or different types of datasets.

Author Response

Thank you for your letter and for the reviewers’ comments concerning our manuscript entitled “Extraction of urban built-up areas based on nighttime lighting data with improved multi-source data” (land-2166783). Those comments are all valuable and very helpful for revising and improving our paper, as well as the important guiding significance to our researches. We have studied comments carefully and have made correction which we hope meet with approval. Revised portion are marked in blue in the paper. The main corrections in the paper and the responds to the reviewer’s comments are as flowing:

  • Point 1: Why is it so important to extract the urban build-up area? I think you can improve the current expression in first graph and provide more evidence(related to SDGs can be considered)..

Response 1: We have re-written this part according to the Reviewer’s suggestion. (line 29-34, in blue)

  • Point 2: I recomment the authors to better construct the Introduction part. You can conclude the previous methods into several catogories and make some comparision. Moreover, I think it might be interesting to introduce more about how previous researches adope POIs/social media/human activities data in this topic(I have read some before).

Response 2: We have re-written this part according to the Reviewer’s suggestion. (line 81-94,line 98-100,line 106-115 in blue).

In the introduction section, we have added many cases, and the overall idea is to firstly summarize the single night light data to extract the built-up area, secondly to combine the night light data with remote sensing data to extract the built-up area from the natural perspective, and thirdly to combine the night light data with social media, points of interest and other big data to extract the built-up area from the economic perspective.

  • Point 3:The references style seems to be wrong in the whole paper! Please check it carefully (page 2 Line 41 for instance).

Response 3: According to the suggestions of reviewers.We have changed the format of all references

  • Point 4:In Method section, you can put a flowchart to better document your experiment.

Response 4: Considering the Reviewer’s suggestion, We put a flowchart. (line 294 ).

  • Point 5: Page 13.It's great that you made comparision among VIIRS, HSI and HP indexes besides your own proposed index. But I think you can provide more details(or figures) to help us understand which method(or index) we should adopt in specific situation.

Response 5: Considering the Reviewer’s suggestion, We added content related to the study topics / content in the introduction. (line 411-412, and line 476-479 in blue).

  • Point 6:In conclusion part, the conclusions(point 1-3) are too long. You can generate some key findings in this part, rather than put too much experimental results.

Response 6: Considering the Reviewer’s suggestion, Based on your suggestions, we have trimmed the conclusion section of the paper. (line 558--582, in blue).

  • Point 7: Please improve your english expression and revised some typos carefully. 

Response 7: Thank you for your suggestions.This paper has been professionally retouched by MDPI's retouching agency and is highlighted in the text

We are very fortunate to have you review our articles as a reviewer and thank you for your rich and valuable comments.Special thanks to you for your good comments. We tried our best to improve the manuscript and made some changes in the manuscript. These changes will not influence the content and framework of the paper.

Round 2

Reviewer 1 Report

The text has been corrected grammatically and stylistically. However, it still contains editorial errors and shortcomings. I encourage the authors to read the article carefully again.

General Comments:

1. Line 41 – reference missing.

2. Line 46 – what means: GB/T50280-98.

3. Line 53, 61, 66, 71 and others – citations problem.

4. Line 186 - .. categories mentioned above ?? These are the data not categories.

5. Table 1 – add column with date.

6. Line 269 - In ENVI  by radiometric calibration of landsat8 data, after atmospheric correction. What does this sentence mean? Landsat (not landsat).

7. Conclusions Section: The conclusions section contains no conclusions. In general, the conclusions should be entirely rewritten. Also, address the ways in which the research may be used in other contexts, and how, more specifically, future work can improve upon the current results.

Author Response

Thank you for your comments on our paper, the relevant responses are in the attached document

Reviewer 2 Report

The authors have addressed all issues I have mentioned. I think the manuscript can be accepted for publication now.

Author Response

Thank you for your comments on our paper

Reviewer 3 Report

The authors have addressed my comments precisely. Consequently, the manuscript has been quite improved and it can be published in its present form.

Author Response

(The authors gave the same response as above.)
